# Brief Communications: Stream Microbes Preferentially Respire Young Carbon within the Ancient Glacier Dissolved Organic Carbon Pool

Amy D. Holt[*1,2], Jason B. Fellman[2], Anne M. Kellerman[1], Eran Hood[2], Samantha H. Bosman[1], Amy M. McKenna[3,4], Jeffery P. Chanton[1] & Robert G. M. Spencer[1]

[1]National High Magnetic Field Laboratory Geochemistry Group and Department of Earth, Ocean, and Atmospheric Science, Florida State University, Tallahassee, FL 32306, USA.
[2]Program on the Environment and Alaska Coastal Rainforest Center, University of Alaska Southeast, Juneau, AK, 99801, USA.
[3]Ion Cyclotron Resonance Facility, National High Magnetic Field Laboratory, Florida State University, Tallahassee, FL 32310-4005, USA
[4]Department of Soil Crop Sciences, Colorado State University, Fort Collins, Colorado 80523-1170, United States

Correspondence to: Amy D. Holt (adh19d@fsu.edu)

**Short Summary.** Glacier runoff is a source of old, bioavailable dissolved organic carbon (DOC) to downstream ecosystems. The DOC pool is composed of material of various origin, chemical character, age and bioavailability. Using bioincubation experiments we show glacier DOC respiration is driven by a young source, rather than ancient material which comprises the majority of the glacier carbon pool. This young, bioavailable fraction could currently be a critical carbon subsidy for recipient food webs.

**Abstract.** Glaciers export ancient, bioavailable dissolved organic carbon (DOC). Yet, the sources of organic carbon (OC) underpinning bioavailability are poorly constrained. We assessed the isotopic composition of respired OC from bioincubations of glacier DOC. Relative to bulk DOC, respired OC was younger (+4,350 – 8,940 years) and $^{13}C$ enriched (+9.2 – 12.2 ‰), consistent with utilization of an in situ produced microbial carbon source. These findings provide direct evidence that a hidden pool of young OC may underpin the high bioavailability of ancient glacier DOC.

## 1 Introduction

Glacier dissolved organic carbon (DOC) has been characterized as ancient and highly bioavailable to aquatic microbes (Hood et al., 2009). Hence, glacier runoff is thought to stimulate downstream heterotrophy, and ultimately release relic carbon to the atmosphere (Hood et al., 2009). The glacier DOC pool is derived from a mixture of organic carbon (OC) of various provenance, chemical composition, and age, including anthropogenic aerosols, soil and plant-derived material (subglacial or windblown), as well as in situ microbial production either on or beneath the glacier (Hood et al., 2009; Stubbins et al., 2012; Spencer et al.,

2014; Musilova et al., 2017; Smith et al., 2017; Behnke et al., 2020, Holt et al., 2021, 2023, 2024). However, it remains unclear which OC sources are responsible for the high bioavailability of glacier DOC. Elucidating the source of bioavailable organics is essential for understanding the fate of glacier-derived DOC and how this pool may change with glacier recession.

Observations of the source of the bioavailable fraction of glacier DOC are ambiguous, since past work suggests that either an aged, or a young component and source of DOC may be most bioavailable. Macroinvertebrates in glacier-fed streams and forelands have been found to be [14]C depleted (i.e., old; Hågvar and Ohlson, 2013; Fellman et al., 2015), indicating that aged OC is assimilated into food webs and thus perhaps underpins glacier DOC bioavailability. Furthermore, the percentage of bioavailable DOC has been negatively correlated with $\Delta^{14}$C-DOC suggesting that aged DOC may be most bioavailable (e.g., Hood et al., 2009). However, this relationship has often been observed in watersheds covering broad gradients of glacier influence, where inputs of non-glacial DOC confound precise identification of the source(s) of bioavailable OC within the glacier DOC pool. Recent molecular-level assessment of supraglacial and outflow dissolved organic matter (DOM) composition has shown that the relative abundance (RA) of bioavailable, aliphatic compounds increases as the DOC pool becomes younger and $\delta^{13}$C enriched (Holt et al., 2023, 2024), further complicating whether the aged component of DOC contributes to the bioavailable fraction. Similarly, the concentration of bioavailable compounds (e.g., carbohydrates and amino acids) has been shown to increase with in situ microbial OC production on the glacier surface (Musilova et al., 2017). Together, these recent studies suggest that young, in situ-derived OC could underpin the high bioavailability of DOC in supraglacial ecosystems and glacier outflows.

Here we investigate the age and potential sources of the respired fraction of DOC in a supraglacial stream and three glacier outflows in the Alaska Coast Mountains (Figure 1). We quantified the carbon isotopic ($\delta^{13}$C and $\Delta^{14}$C - i.e., source and age) signature of bulk glacier DOC and, for the first time, respired OC (as $CO_2$) using respiratory carbon recovery system (RCRS) experiments, which allow the $CO_2$ produced by microbial respiration of DOC to be captured and its isotopic signature ($\delta^{13}$C-$CO_2$ and $\Delta^{14}$C-$CO_2$) assessed (McCallister et al., 2006). Isotopic signatures were used in conjunction with molecular-level data derived from 21 T Fourier transform ion cyclotron resonance mass spectrometry (FT-ICR MS) to evaluate the interplay between DOM composition and the age and source of respired OC. We hypothesized that respiratory $CO_2$ would be isotopically younger and [13]C enriched compared to bulk DOC, consistent with the notion that in situ microbial production within glacier ecosystems fuels microbial heterotrophy (e.g., Musilova et al., 2017; Smith et al., 2017; McCrimmon et al., 2018). Ultimately, our findings provide novel insights into glacier DOC source and bioavailability, with ramifications for our understanding of how the OC subsidy glaciers provide to downstream ecosystems may be altered by continued glacier retreat.

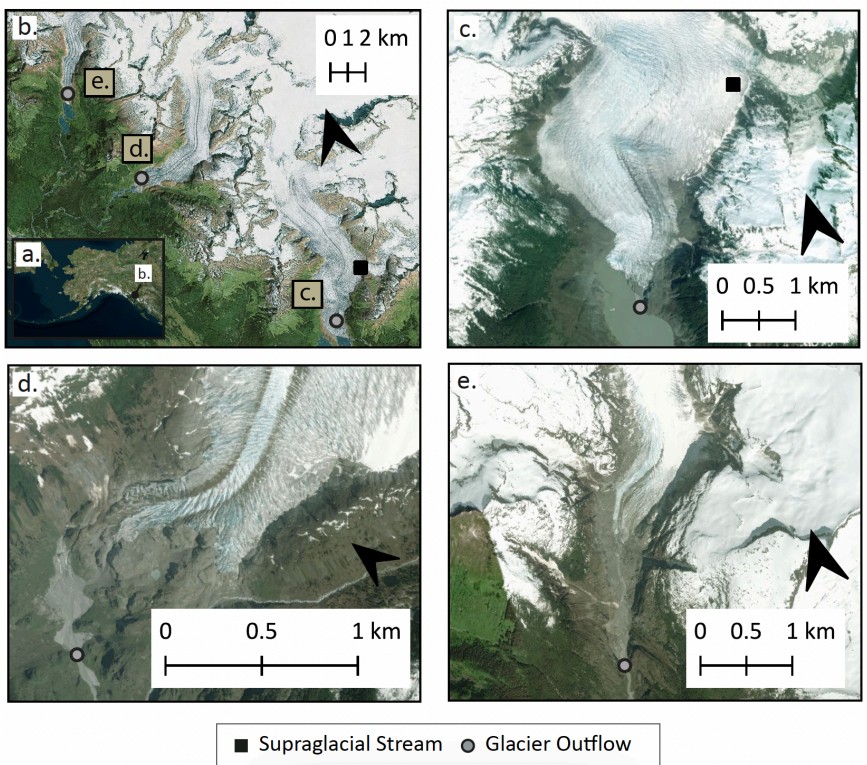

**Figure 1:** Location of July 2022 sample sites within (a.) North America and (b.) the Juneau Icefield, highlighting panels (b.-d.) and (b.) Mendenhall Glacier, (c.) Herbert Glacier and Eagle Glacier catchments. Map data: Bing Satellite.

## 2 Materials and Methods

### 2.1 Study Sites and Sample Collection

Water samples were collected between the 11th and 14th of July 2022, from three glacierized watersheds in coastal southeast Alaska (Figure 1). The study area is situated within the Juneau Icefield, in the coastal temperate rainforest and has a cool (annual mean temperature 5.6°C), maritime climate, with the majority of precipitation falling in autumn and winter (Behnke et al., 2020). The geology of the upper watersheds, where the glaciers are found, is dominated by Tertiary-Cretaceous aged, foliated tonalite sill of the coast plutonic complex (Wilson et al., 2015). These glaciers are well studied and are known to have discharge regimes and biogeochemical characteristics representative of glacial systems throughout the Gulf of Alaska. These glacier rivers are highly turbid, with low summer temperatures (<5°C) and oligotrophic conditions (Hood and Scott, 2008; Fellman et al., 2014; Spencer et al., 2014). Based on past observations of conductivity, at the time of sampling, water residence times within the glaciers are short (~hours), and subglacial drainage is relatively efficient, with the vast majority of meltwater having a supraglacial origin (Spencer et al., 2014).

Water samples were collected from the surface and outflow of Mendenhall Glacier, as well as downstream (~ ≤1 km) of the terminus of both Eagle and Herbert Glaciers. Between the glacier terminus and sampling sites glacier outflow rivers flowed through recently deglaciated terrain (i.e., barren ground of cobble, gravel and glacier silt, with few colonizer plants) and thus there was limited potential for OC inputs from vascular plants and soil organic matter. At Mendenhall, outflow sampling was conducted on a rock/silt bar <100 m from the glacier outflow. Water here is extremely turbulent and flows rapidly into Mendenhall lake. As such, there is limited influence of the lake water, and sampling is representative of water exiting the glacier. The supraglacial sample was collected from a small (<1 m across) flowing supraglacial stream on the bare ice surface accessed by helicopter, ~ 3 km upslope from the glacier terminus.

At each site, water was immediately filtered to 0.45 μm (Geotech Polyethersulfone dispos-a-filter™ capsule), acidified to pH 2 (10 M HCl) and stored (<2 weeks) at -20°C in the dark until further processing. Filtrate was collected for RCRS experiments, and analysis of DOC concentration, carbon isotopes of DOC and molecular-level composition. Samples were stored in 125 mL (DOC) or 1 L (other analyses) polycarbonate bottles. Additionally, at each site, water was also filtered to 1.6 μm using pre-combusted GF/A filters, stored in a 500 mL polycarbonate bottle at 4°C in the dark, and used in the preparation of the RCRS experiment inocula.

**2.2 Respiratory Carbon Recovery System Experiments**

The RCRS experiments were conducted following established methodology (McCallister et al., 2006). For each site, 0.45 μm filtrate was decanted into a 1 L acid-washed, pre-combusted serum bottle and then crimp-sealed, ensuring the bottle was gas-tight. Samples were then sparged in the dark with ultra-high purity He (2 h, 0.08 L min$^{-1}$) to strip dissolved inorganic carbon (DIC) from solution. Subsequently, samples were neutralized (~pH 7) with DIC-free NaOH, and reoxygenated with ultra-high purity O$_2$ (0.5 h, 0.08 L min$^{-1}$) until >20.95% O$_2$ air saturation as monitored by a PreSen Fibox O$_2$ needle probe.

Inocula were prepared from the 1.6 μm filtrate. Aliquots from each site were mixed in equal proportion to form a composite inocula, an approach used in laboratory bioassays because it controls for the potential influence of site-specific differences in bacterial community composition on DOC removal. For each experiment, 42 mL of composite inocula was filtered through a 0.2 μm Whatman® polycarbonate Nuclepore Track-EtchMembrane filter using a pre-cleaned glass filter tower. Using a needle and syringe, DIC-free incubation water was extracted from the serum bottle and used to resuspend microbes harvested on the filter. This process acts to limit the amount of DIC entering the DIC-free, closed-system experiment. The resuspended microbial community (~2.5 mL) was injected into the serum bottle. Samples were incubated at 22°C in the dark for 28 days, as is standard for past DOC and glacier DOC bioincubations (e.g., Hood et al., 2009; Holt et al., 2023).

Following incubation, samples were acidified to pH 2 using DIC-free HCl. Evolved CO$_2$ from microbial respiration of DOC was sparged from solution (2 h, 0.08 L min$^{-1}$) with ultra-high purity He and trapped cryogenically (liquid N$_2$) on a vacuum

line. The $CO_2$ was then purified through a series of cryogenic traps, before being quantified and isolated in a pre-combusted (550°C, 5h) break seal (McCallister et al., 2006).

**2.3 Dissolved Organic Carbon Quantification and Carbon Isotopic Analyses**

Concentrations of DOC were measured on a Shimadzu TOC-L$_{CPH}$ analyzer following standard methods (e.g., Holt et al., 2024 and references therein). Before analysis, samples were sparged with air for five minutes at a flow rate of 0.08 L min$^{-1}$ to remove DIC from solution. Measured concentrations are based on 3 of 7 replicate injections with a coefficient of variance of <2%. $^{13}$C and $^{14}$C were measured via isotope ratio mass spectrometry (IRMS) and accelerator mass spectrometry, respectively at Woods Hole Oceanographic Institution. For DOC isotopes, samples were UV-oxidized, and the resultant $CO_2$ cryogenically trapped for analysis. $\delta^{13}$C values measured by IRMS have a typical precision of <0.2‰ (Xu et al., 2021). Estimates of the contributions from radiocarbon dead (-1000 ‰) versus modern (95 % of $^{14}$C concentration in 1950 of NBS Oxalic Acid I normalized to $\delta^{13}$C$_{VPDB}$= -19 ‰) OC were calculated from fraction modern ($F_m$) values (Table 1), where the percentage of radiocarbon dead material was determined as 1-$F_m$ (Stubbins et al., 2012). Measurement error on $F_m$ ranged from 0.0017 – 0.0025 (Table 1), making little quantitative difference to calculated values (i.e., ‰ and yBP) and estimated source contributions.

**2.4 Molecular-level Analysis of Dissolved Organic Matter**

Water samples were solid phase extracted (100 mg Bond Elut PPL cartridges) and analyzed via negative-ion electrospray ionization 21 T FT-ICR MS using standard methods (e.g., Holt et al., 2021, 2023, 2024 and references therein). In brief, the volume extracted was adjusted dependent on the sample DOC concentration to achieve a target eluent concentration of 40 µg C L$^{-1}$. Cartridges were eluted with 1 mL of methanol. Mass spectra were formed from 100 scans conditionally co-added and phase corrected. Spectra were internally calibrated in Predator analysis using the 'walking calibration'. Peaks with greater than the baseline signal-to-noise plus 6$\sigma$ were exported to a peak list. Elemental composition was assigned to peaks within the bounds $C_{1-100}H_{4-200}O_{1-30}N_{0-4}S_{0-2}$ (error $\leq \pm$ 0.3 ppm) using PetroOrg©. Assigned molecular formulae were classed by heteroatom content (CHO, CHON, CHOS and CHONS) and grouped into commonly used, operational compound classes using the modified aromaticity index (AI$_{mod}$) and elemental stoichiometry (Holt et al., 2021 and references therein): condensed aromatics and polyphenolic (AI$_{mod}$ values >0.67 and of 0.5–0.67, respectively), highly unsaturated and phenolic (HUP; AI$_{mod}$ of <0.5 and H/C < 1.5) and aliphatics (H/C $\geq$ 1.5 and O/C $\leq$ 0.9). Sugar-like compounds (H/C $\geq$ 1.5 and O/C > 0.9) were also identified making up $\leq$ 0.2 %RA of all samples and thus are not discussed further.

**Table 1:** Dissolved organic carbon (DOC) concentrations, and carbon isotopic signatures of DOC and respiratory $CO_2$, along with the offset between measured carbon isotopic values.

| | Eagle Glacier Outflow | Herbert Glacier Outflow | Mendenhall Glacier Outflow | Mendenhall Supraglacial Stream |
|---|---|---|---|---|
| DOC (mg C $L^{-1}$) | 0.4 | 0.7 | 0.4 | 0.6 |
| $\delta^{13}$C-DOC (‰) | -27.8 | -28.7 | -28.8 | -27.7 |
| $\delta^{13}$C-$CO_2$ (‰) | -18.3 | -19.5 | -19.2 | -15.5 |
| $\delta^{13}$C-offset (‰) | 9.5 | 9.2 | 9.6 | 12.2 |
| $\Delta^{14}$C-DOC ($F_m$) | 0.34 ± 0.0018 | 0.26 ± 0.0017 | 0.45 ± 0.0017 | 0.41± 0.0018 |
| $\Delta^{14}$C-$CO_2$ ($F_m$) | 0.86 ± 0.0018 | 0.79 ± 0.0025 | 0.77 ± 0.0018 | 0.98 ± 0.0020 |
| $\Delta^{14}$C-DOC (‰; yBP) | -667.5 (8,780) | -741.7 (10,800) | -553.4 (6,410) | -589.1 (7,080) |
| $\Delta^{14}$C-$CO_2$ (‰; yBP) | -146.2 (1,200) | -213.1 (1,860) | -232.9 (2,060) | -30.8 (180) |
| $\Delta^{14}$C-offset (‰) | 521.3 | 528.6 | 320.5 | 558.3 |

### 3 Results and Discussion

**3.1 Carbon Isotopic and Molecular Composition of Glacier Dissolved Organic Matter**

Concentrations of DOC ranged from 0.4 – 0.7 mg C $L^{-1}$, consistent with published values for these sites (Table 1; Hood et al., 2009; Stubbins et al., 2012; Spencer et al., 2014; Behnke et al., 2020). Supraglacial stream and outflow DOC was ancient (median $\Delta^{14}$C-DOC -628.3 ‰, range 10,800 – 6,410 yBP; Table 1, Figure 2) as is typical for DOC derived from these and other glacier ecosystems (Hood et al., 2009; Stubbins et al., 2012; Holt et al., 2024). There was minimal variability in $\delta^{13}$C-DOC values, which ranged from -28.8 to -27.7 ‰ (Table 1, Figure 2), overlapping with those reported previously for glacier outflow DOC (e.g., Hood et al., 2009; Holt et al., 2024).

At the molecular-level, all sites were dominated by CHO-only (74.6 – 79.5 %RA) and HUP (58.4 – 73.0 %RA) formulae, as is typical for DOM in glacial and non-glacial aquatic ecosystems (Supplementary Table 1, Figure 3; e.g., Behnke et al., 2020). Heteroatom containing (CHON, CHOS, and CHONS) and aliphatic formulae were abundant in the supraglacial stream and outflows (median 24.3 and 22.1 %RA, respectively), and condensed aromatic and polyphenolic compounds were a minor fraction of DOM composition (5.9 – 8.7 %RA; Supplementary Table 1, Figure 3). This is consistent with previous molecular-level assessments from these sites and other glacier ecosystems, where glacier DOM is characterized as relatively heteroatom-enriched and low in aromaticity compared to rivers with no to low glacier inputs (e.g., Behnke et al., 2020; Holt et al., 2024).

### 3.2 Composition and Source of Glacier Dissolved Organic Matter

The isotopic and molecular-level composition of the supraglacial stream and glacier outflows was consistent with OC derived from a mixture of sources. Past studies of southeast Alaskan glaciers suggest that a substantial fraction of glacier DOC is derived from anthropogenic aerosols (Stubbins et al., 2012, Spencer et al., 2014; Holt et al., 2024). Based on simple mixing of radiocarbon dead (i.e., a purely fossil fuel source) and modern OC, ~55 – 74 % of DOC across our study sites could have been derived from fossil fuel combustion byproducts, in line with past estimates (Table 1; Stubbins et al., 2012). This material was consistent with the presence of condensed aromatic compounds on the glacier surface and in the outflows, and may have also contributed to the observed aliphatic-rich composition, especially if photodegraded (Supplementary Table 1, Figure 3; Holt et al., 2021). In glacier outflows, a component of aged OC may also originate from subglacial material (e.g., overridden soils enriched in aromatic moieties), or DOC aged during glacier ice formation (Hood et al., 2009; Stubbins et al., 2012). Nonetheless, given the compositional similarity between supraglacial and outflow DOM (Figure 2 & 3) and the residence time of glacier ice in these catchments (~300 yBP; Stubbins et al., 2012) these contributions are likely marginal.

A sizeable portion of DOC was also derived from younger or modern (up to 45 % of DOC) OC sources, rich in aliphatic moieties. This likely included in situ microbial production (e.g., by ice algae or cyanobacteria) on the glacier surface, which is known to produce aliphatic compounds (Musilova et al., 2017), and potentially vegetation and soil organic matter. Soil and vegetation OC could have been sourced from lateral inputs, or atmospheric deposition (Spencer et al., 2014; Holt et al., 2024). This organic matter may have been the source of the relatively small fraction of polyphenolic compounds (<7.0 %RA) within the glacier DOM pools (Supplementary Table 1, Figure 3) and potentially saturated compounds particularly if photodegraded (Holt et al., 2021).

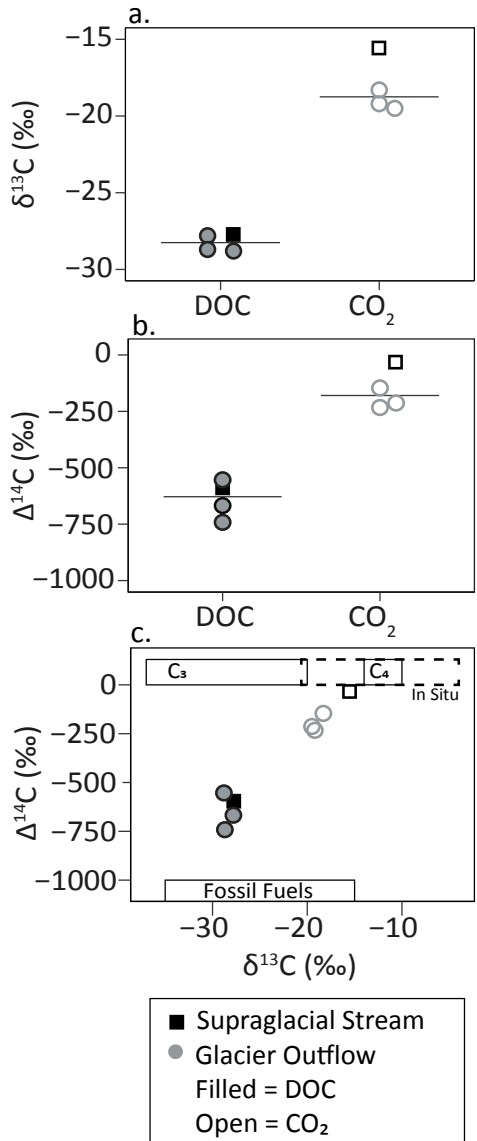

**Figure 2:** The (a.) $\delta^{13}C$ and (b.) $\Delta^{14}C$ signature of DOC and respiratory $CO_2$, and (c.) the respective signatures in relation to possible endmember sources. Samples are colored and shaped according to their type (outflow and supraglacial – grey circle and black square, respectively). Filled and open symbols represent DOC and $CO_2$, respectively. (a. and b.) The solid line represents the median value. (c.) Boxes represent endmember sources: $C_3$ plants (Kohn, 2010), $C_4$ plants (Cerling et al., 1997), fossil fuels (Wang et al., 2022), in situ production taken from the range published for cyanobacteria (Schmidt et al., 2022) and sea ice algae (Hobson and Welch, 1992; McMahon et al., 2006). A description of $C_3$ and $C_4$ plants is found in the supplemental.

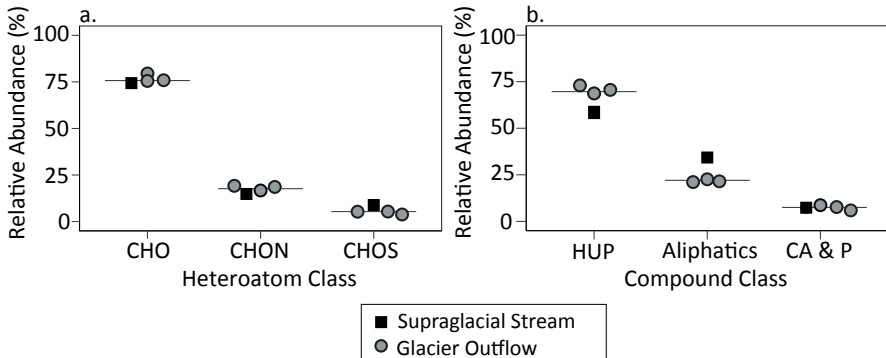

**Figure 3:** Molecular-level composition of dissolved organic matter (DOM) before bioincubation. Plots show the percent relative abundance (RA) of: (a.) heteroatom and (b.) compound classes. Samples are colored and shaped according to type (outflow and supraglacial - grey circle and black square, respectively). The solid line represents the median value. HUP: highly unsaturated and phenolic. CA & P: condensed aromatic and polyphenolic.

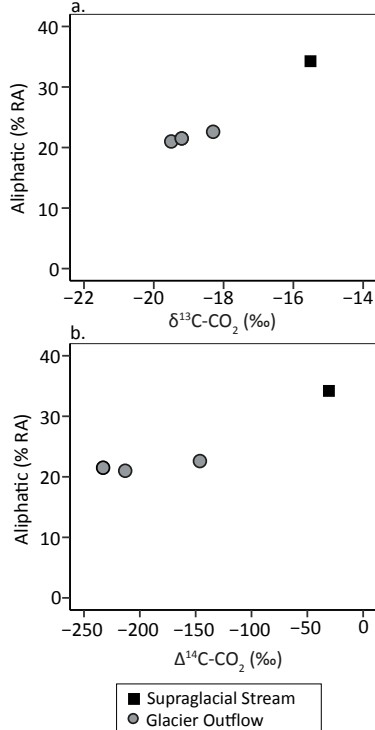

**Figure 4**: Association between carbon isotopic signature of respiratory $CO_2$ and the relative abundance (RA) of aliphatic compounds in the DOM pool (a. and b. $\delta^{13}C$-$CO_2$ and $\Delta^{14}C$-$CO_2$, respectively). Samples are colored and shaped according to type (outflow and supraglacial – grey circle and black square, respectively).

### 3.3 Isotopic Composition of Respiratory Carbon and Relation to Molecular Composition of Dissolved Organic Matter

The RCRS experiments using supraglacial and glacier outflow DOC produced respiratory $CO_2$ (0.2 - 0.3 mgC; Supplementary Table 2) with $\delta^{13}C$ and $\Delta^{14}C$ values ranging from -19.5 to -15.5 ‰ and -232.9 to -30.8 ‰ (2,060 – 180 yBP), respectively (Table 1). These values were positively offset ($\delta^{13}C$ and $\Delta^{14}C$-$CO_2$ median +9.6 ‰ and +525.0 ‰, respectively) from the isotopic values of the initial DOC pool (Figure 2). Hence, during the incubation, microbial community respiration was predominately fueled by organic compounds within the DOC pool that were relatively enriched in $^{13}C$ and $^{14}C$ (i.e., younger) compared to bulk DOC. Enriched $\delta^{13}C$-$CO_2$ values may be associated with an increased %RA of aliphatic compounds in the initial DOM pool, and in the case of the supraglacial stream appears to coincide with substantially younger $CO_2$ (Figure 4). Given the small dataset ($n = 4$), it is unclear whether this represents a compositional trend. However, the data suggest that the $^{13}C$ and $^{14}C$ enriched source respired during the incubations could contain a greater relative abundance of aliphatic compounds.

### 3.4 Sources of Respired Glacier Dissolved Organic Carbon

The RCRS experiments demonstrated that microbial respiration was fueled by younger and $^{13}C$ enriched OC compared to the bulk DOC pool (Figure 2). Although isotopic signatures of OC endmembers are poorly constrained for glaciers, the $\delta^{13}C$ values of respiratory $CO_2$ (-19.5 to -15.5 ‰) overlapped with those of sea ice algae (-21 to -16 ‰) and cyanobacteria (-16 to -6 ‰), providing a line of evidence to suggest that bioavailable OC could be derived from in situ microbial sources (Table 1, Figure 2; Hobson and Welch, 1992; McMahon et al., 2006; Schmidt et al., 2022). Similarly, total OC concentrations of cryoconite hole sediments have been shown to positively correlate with $\delta^{13}C$, where enriched values (~-16 to -10 ‰) were postulated to reflect greater relative contributions of in situ produced OC compared to atmospherically deposited OC (Schmidt et al., 2022). Since microbial production on the glacier surface would be associated with enriched $\delta^{13}C$ signatures (e.g., Schmidt et al., 2022; Holt et al., 2024), the respired $^{13}C$ enriched OC that we observed may be largely derived from in situ production (e.g., snow, ice algae or cyanobacteria). Furthermore, the supraglacial stream had the most aliphatic-rich DOM and the most $^{13}C$ and $^{14}C$ enriched $CO_2$ (Figure 2, 3 and 4). This is consistent with autochthonously sourced aliphatic organic matter (Musilova et al., 2017; Holt et al., 2024), and further suggests that respired organics were derived from in situ production. Additionally, the relative similarity in molecular and isotopic composition (DOC and respiratory $CO_2$) between the supraglacial stream and the glacier outflow samples suggests that a portion of the bioavailable organic carbon exported from these glaciers originates from recent in situ production on the surface rather than subglacial sources (Table 1, Figures 2 & 3). However, subglacial sources such as paleosols and microbial chemotrophic processes could still contribute organic carbon to the outflow DOM pool. Our findings are in line with past studies highlighting that microbes in cryoconite hole sediments fix atmospheric $CO_2$, and recently fixed OC compounds on glacier surfaces support microbial heterotrophy (Musilova et al., 2017; Smith et al., 2017; McCrimmon et al., 2018). It is conceivable that an alternative source of young, $^{13}C$ enriched OC could have fueled respiration. This could include modern $C_4$ plant material ($\delta^{13}C$ -14 to -10 ‰; see supplementary for description of $C_3$ and $C_4$ plants), for

example aliphatic moieties from combustion byproducts or pollen (Figure 2; Cerling et al., 1997; Holt et al., 2021, 2023). Nonetheless, the dominance of $C_3$ vegetation ($\delta^{13}$C -37 to -20 ‰; Khon., 2010) in the coastal temperate rainforest of southeast Alaska makes a $C_4$ source unlikely and instead supports in situ microbial production on the glacier surface as an important source of young, bioavailable OC to the glacier DOC pools.

Despite overall shifts to younger isotopic signatures relative to bulk glacier DOC (median offset +525.0 ‰), the evolved $CO_2$ produced from microbial respiration was still slightly aged (-232.9 to -30.8 ‰, 2,060 to 180 yBP; Table 1, Figure 2). This aged OC could have been sourced from contributions from radiocarbon dead material like fossil fuel combustion byproducts. Fossil fuel sources and their photodegraded byproducts are known to contain aliphatic compounds (Holt et al., 2021) and therefore could be a source of bioavailable OC respired during the incubation period. Fossil fuel sources are on average more $^{13}$C depleted compared to the isotopic composition of respiratory $CO_2$ (Table 1; Wang et al., 2022). Therefore, if fossil fuels contribute $^{13}$C-enriched OC, this could have originated from a $^{13}$C enriched fossil fuel source, compounds within the sources' OC pool that are relatively $^{13}$C enriched compared to the bulk, or result from photodegradation of fossil fuel-derived compounds which is known to result in $^{13}$C enrichment (Spencer et al., 2009). In outflows, ancient DOC compounds could also be derived from aged ice-locked, and subglacial sources (e.g., paleosols and microbial chemotrophic processes), although compositional and carbon isotopic evidence to support this conclusively is yet to be observed for Alaskan glaciers (Stubbins et al., 2012; Spencer et al., 2014). The combined $^{13}$C enriched and relatively young age of respiratory $CO_2$ (Table 1), confirms that any aged, $^{13}$C depleted source likely made up a small fraction of respired DOC during the incubations. It remains unclear how stream microbes utilize different OC compounds within the 28-day incubation period, and whether or not different sources, including ancient OC, are preferentially metabolized or potentially prime respiration of less favorable OC sources. Additionally, since the samples were collected during peak melt (July) from a small number ($n$=3) of proximate glaciers in southeast Alaska, it is yet to be quantified if temporal shifts and regional differences in bulk DOM composition (Spencer et al., 2014; Holt et al., 2024) impact the source and magnitude of respired OC. Given these bioincubation experiments were conducted under uniform conditions (i.e., pH, temperature and $O_2$ saturation), comparison of data between them is appropriate, but it is unclear how in situ physiochemical properties and geochemistry may effect DOC metabolism and fate in the real world. Moreover, microbial carbon utilization and DOC bioavailability in the environment are a function of both respiration and biomass production, the latter of which was not assessed here. Ultimately, DOC from glacier ecosystems globally is exported to a range of aquatic environments (e.g., proglacial lakes, lower reaches of rivers, fjords, estuaries and near shore marine environments) with variable transit times, residence times, and environmental conditions. How these variances effect which glacier OC sources are used by the microbial food web is yet to be determined. That said, this study provides the first quantitative glimpse into the carbon sources underpinning the bioavailability of the ancient glacier DOC pool using a direct experimental approach.

## 4    Summary and Implications

Glaciers globally have been described as a source of ancient, bioavailable DOC to downstream and marine food webs (e.g., Hood et al., 2009; Fellman et al., 2015; Holt et al., 2024). Our study in the Alaska Coast Mountains provides direct evidence that despite being aged, the high bioavailability of glacier DOC may be predominantly underpinned by younger OC, likely sourced from in situ microbial production on the glacier surface. Though yet to be quantified, this in situ source of carbon could be critical for stream heterotrophy across deglaciating watersheds and could currently subsidize food webs that support socioeconomically important fisheries (e.g., in the Gulf of Alaska, and broadly the northern Pacific and Atlantic). Despite the dominance of young OC, we show respiratory $CO_2$ was slightly aged (180 – 2,060 yBP) across the study sites demonstrating that glacier runoff mobilizes some relic carbon that can be assimilated into food webs and released to the atmosphere (Hood et al., 2009; Fellman et al., 2015). Nonetheless, if predominately modern, in situ production underpins OC bioavailability across glaciers, as suggested by this dataset, then microbial metabolism of glacier-derived DOC in downstream ecosystems globally may be primarily cycling contemporary material, rather than largely a microbially-mediated release of aged carbon to the atmosphere. Importantly, as glaciers continue to recede, the glacier-derived DOC flux declines and stream physicochemical conditions become more conducive to microbial production (e.g., Hood et al., 2015; Kohler et al., 2024), the source of this bioavailable, modern OC will likely switch from glacier-derived (e.g., glacier surface algal production) towards instream sources.

**Acknowledgements:**

JBF and EH were supported by National Science Foundation (NSF) Alaska EPSCoR Program (OIA-1757348) and Division of Earth Sciences (EAR-2227821). The 21 tesla FT-ICR MS analysis was performed in the Ion Cyclotron Resonance User Facility, National High Magnetic Field Laboratory, Florida, USA, which is supported by the NSF Division of Chemistry and Division of Materials Research through DMR 16-44779 and DMR 2128556. ADH and RGMS are extremely grateful to the Winchester Foundation and the International Association of Geochemistry for research support. NSF supported this research through an INTERN award to ADH through the Arctic Great Rivers Observatory award (RGMS: 1914081).

**Competing Interests:**

The authors declare no competing interests.

**Data Availability:**

Raw FT-ICR MS spectra files, calibrated peak lists, and assigned elemental composition data are available in the Open Science Framework (OSF; https://osf.io/4m2kx/) repository under the following DOI: 10.17605/OSF.IO/4M2KX.

**Author Contribution:**

RGMS research conceptualization; ADH, JBF, EH and RGMS funding acquisition and/or resources; ADH, JBF and EH conducted fieldwork; ADH performed RCRS experiments with support from AMK, SHB and JPC; ADH and AMM performed laboratory analyses; ADH data analysis and interpretation with support and supervision from JBF, EH and RGMS; ADH wrote the original draft and all authors contributed to writing review and editing with significant input from JBF, EH and RGMS.

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
