# Peer review of "Brief Communications: Stream Microbes Preferentially Respire Young Carbon within the Ancient Glacier Dissolved Organic Carbon Pool"

_EGUsphere, 2024_

## Author Comment (AC1)

**FLORIDA STATE UNIVERSITY**

College of Arts & Sciences
*Department of Earth, Ocean & Atmospheric Science*

April 2nd, 2025

Dr. Amy Holt

Earth, Ocean, and Atmospheric Science (EOAS)

Florida State University

Tallahassee, FL 32304

Dear Reviewer 1,

**Brief Communications: Stream Microbes Preferentially Utilize Young Carbon within the Ancient Bulk Glacier Dissolved Organic Carbon Pool.**

Amy D. Holt, Jason B. Fellman, Anne M. Kellerman, Eran Hood, Samantha H. Bosman, Amy M. McKenna, Jeffery P. Chanton & Robert G. M. Spencer.

We sincerely thank you for your positive feedback and helpful comments on our manuscript. Please find below a detailed response to your comments with proposed revisions. Your comments are in italic and our response in regular text. Line numbers refer to the original posted manuscript. We hope with these refinements and additions you now find the paper suitable for publication in The Cryosphere: Brief Communications.

**Response:**

*Although glaciers have previously been shown to export pre-aged DOC that is highly bioavailable. The actual age of glacier DOC consumed by microbial communities has not been identified. This is significant as it has direct implications for atmospheric carbon budgets and positive feedbacks to climate warming.*

We thank you for your positive comments and agree that this work has novel and important implications for our understanding of carbon dynamics and climate warming.

*Abstract: Add "bulk" before…."DOC", to read "Relative to bulk DOC, ..." Remove the "+" before 4,350*

We thank you for the suggestions and agree that 'bulk' should be added to the text.

We would like to keep the '+' in front of the yearly values as this refers to the magnitude of positive offset rather than the age of the $CO_2$ (i.e., those reported in Table 1). We would, however, remove 'BP'.

The sentence would read:

"Relative to **bulk** DOC, respired OC was younger (+4,350 – 8,940 **years**) and $^{13}$C enriched (+9.2 – 12.2 ‰), consistent with utilization of an in situ produced microbial carbon source."

*Introduction: Line 48- The phrasing of "…whether the aged component of glacier DOC is responsible for its high bioavailability" should be changed as it implies that the age determines its bioavailability, which is not the case as other chemical, physical and biological parameters determine the portion consumed. Perhaps something along the lines of "whether the aged component contributes to the bioavailable fraction".*

Thank you for catching this and we agree that this could be misleading to the reader. We would revise the text as you suggested to "Recent molecular-level assessment of supraglacial and outflow dissolved organic matter (DOM) composition has shown that the relative abundance (RA) of bioavailable, aliphatic compounds increases as the DOC pool becomes younger (Holt et al., 2023; Holt et al., 2024), further complicating whether the **aged component of DOC contributes to the bioavailable fraction**."

*Line 60- delete "bioavailable" after "…the age and source of respired, bioavailable OC"*

Thank you for the suggestion. The sentence would read: "Isotopic signatures were used in conjunction with molecular-level data derived from 21 T Fourier transform ion cyclotron resonance mass spectrometry (FT-ICR MS) to evaluate the interplay between DOM composition and the age and source of respired OC."

*Methods: Line 75-82- How long was the filtrate stored prior to incubations? What was the filtrate stored in and what volume? The filtrate for respiration experiments was acidified and frozen? Did this result in flocculation of DOC when thawed for these incubations and measurements? If so what fraction was lost as flocculant?*

We thank you for your questions. To clarify, all 0.45 µm filtrate was acidified to pH 2 and stored frozen (lines 78-80). The filtrate was stored in 1L polycarbonate bottles for FT-ICR MS, bulk isotopic and RCRS incubations, and in 125 mL polycarbonate bottles for DOC concentration. For the inocula (1.2 µm), filtrate was stored in 500 mL polycarbonate bottles (4°C). Filtrate was stored for <2 weeks before subsequent processing, incubation set up or analysis. We did not note any flocculation upon thaw of the 0.45 µm samples. This is unsurprising given the supraglacial and glacier outflow DOM pool is known to be low in aromaticity (also shown here in section 3.3; Holt et al., 2024, Kellerman et al., 2021), and it is these aromatic compounds that are prone to flocculation (e.g., Fellman et al., 2008).

We suggest amending the text to recognize the timeframe of storage and bottles used.

The text would read:

"At each site, streamwater was immediately filtered to 0.45 µm (Geotech Polyethersulfone dispos-a-filter™ capsule), acidified to pH 2 **(10 M HCl)** and stored **(<2 weeks)** at -20°C in the dark until further processing. Filtrate was collected for RCRS experiments, and analysis of DOC concentration, carbon isotopes of DOC and molecular-level composition. Samples were **stored in 125 mL (DOC) or 1 L (other analyses) acid-washed (10% HCl v/v) polycarbonate bottles**. Additionally, at each site, streamwater was also filtered to 1.6 µm using pre-combusted GF/A filters, stored **in a 500 mL acid-washed polycarbonate bottle** at 4°C in the dark and used in the preparation of the RCRS experiment inocula."

*Line 87- How was the initial stripping of DIC from the incubation water verified? Was DIC or pCO2 measured to assure complete removal?*

Thank you for your question. DIC stripping was not directly examined here. However, our past testing of the RCRS method has shown near complete DIC removal (to < 0.04 mg C L$^{-1}$) on standards with initial DIC concentrations of ~80 mg C L$^{-1}$ within the first 75 minutes of sparging. Similarly, McCallister, Guillemette, and Del Giorgio (2006), who the RCRS methodological design is based on, showed removal of > 2500 mg C L$^{-1}$ from a 20 L container to ≤0.06 mg C L$^{-1}$ within 80 minutes of sparging at a flow rate of 1 L min$^{-1}$. Given the significantly lower DIC concentration of glacier rivers and watersheds (< 10 mg C L$^{-1}$; Andrews, Jacobson, Osburn, & Flynn, 2018; Harley et al., 2023), the incubations volumes (1L), and the extended sparging time (2 hours), we are confident of complete DIC removal with this method. Furthermore, the sparging time and flow rate (0.08 L min$^{-1}$) used here are in line with other DOC methods that require DIC removal, like DOC quantification via the non-purgeable organic carbon method (NPOC, i.e., the TOC analysis conducted here). For NPOC, typically sample waters are sparged for five minutes at a flow rate of 0.08 L min$^{-1}$ (Stubbins & Dittmar, 2012). These were also the conditions used for DOC analysis in this study. Finally, we did continuously monitor stripping of dissolved gases through a PreSen Fibox O$_2$ needle probe. After 2 hours of sparging, O$_2$ in solution was <0.5 %, also confirming near-complete to complete removal of gases (encompassing CO$_2$).

*Line 88- how supersaturated where the incubations with O2, that is an extremely high rate of O2 flow and duration for a 1L incubation. It would have been ideal to return to initial O2 conditions. Supersaturation of O2 can significantly impair microbial decomposition due to formation of reactive oxygen species, interference with enzymes etc. What were the O2 levels at the time of incubation initiation?*

We can confirm that a far lower rate (0.08 L min$^{-1}$) was used, and 0.8 L min$^{-1}$ was reported in error. Thank you for catching this typographical error. Flow rate was monitored using an in-line Dywer Instrument Flow Meter, and the samples were reoxygenated to a concentration >20.95% O$_2$ monitored by a PreSen Fibox O$_2$ needle probe. This means the incubations were conducted under O$_2$ replete conditions. We propose the following edit:

"Subsequently, samples were neutralized (~pH 7) with DIC-free NaOH, and reoxygenated with ultra-high purity O$_2$ (0.5 h, 0.08 L min$^{-1}$) until >20.95% O$_2$ air saturation as monitored by a PreSen Fibox O$_2$ needle probe."

We appreciate the limitations of bottle-based incubations and experimental conditions on the metabolism of organics, including rates of utilization. We propose to expand our discussion of the limits of our findings to encompass this point.

We suggest the following (lines 230 onwards):

"It remains unclear how stream microbes utilize different OC compounds within the 28-day incubation period, and whether or not different sources, including ancient OC, are preferentially metabolized or potentially prime respiration of less favorable OC sources. Additionally, since the samples were collected during peak melt (July) from a small number (*n*=3) of proximate glaciers in southeast Alaska, it is yet to be quantified if temporal shifts and regional differences in bulk DOM composition (Spencer et al., 2014; Holt et al., 2024) impact the source and magnitude of respired OC. **Given these bioincubation experiments were conducted under uniform conditions (i.e., pH, temperature and O$_2$ saturation), comparison of data between them is appropriate, but it is unclear how in situ physiochemical properties and geochemistry may affect DOC metabolism and fate in the real world**. Moreover, microbial carbon utilization and DOC bioavailability in the environment are a function of both respiration and biomass production, the latter of which was not assessed here. Ultimately, DOC from glacier

ecosystems globally is exported to a range of aquatic environments (e.g., proglacial lakes, lower reaches of rivers, fjords, estuaries and near shore marine environments) with variable transit times, residence times, and environmental conditions. How these variances effect which glacier OC sources are used by the microbial food web is yet to be determined. That said, this study provides the first quantitative glimpse into the carbon sources underpinning the bioavailability of the ancient glacier DOC pool using a direct experimental approach."

*Line 100- How was the evolved CO2 collected prior to transfer to the vacuum line? Cryogenically? The flow rate of 0.8 L min$^{-1}$ is much too fast for quantitative trapping. For example, Beaupre et al. 2007 (L&O methods) used a flow rate of 200ml min$^{-1}$. The 0.8 L min$^{-1}$ means that only a portion of the respired C would be captured, with a significant portion expelled from the Liq N2 trap, resulting in isotopic fractionation. Thus the isotopic values measured do not reflect the signature of respired C, but only whatever fraction which was trapped and recovered. How were these flow rates measured?*

Thank you for your question and for catching the error in the flow rate. We can confirm that a flow rate of 0.08 L min$^{-1}$ was used and monitored throughout using an in-line Dywer Instrument Flow Meter. As clarified above, this flow rate is typical of dissolved organic carbon methods which require DIC removal from solution. We apologize for the error.

To clarify, $CO_2$ was sparged from solution (0.08 L min$^{-1}$) and captured directly on a vacuum line using cryogens. After 2 hours of sparging, the captured $CO_2$ was then purified cryogenically on the same vacuum line. For the Reviewer's benefit, we also note there is an in-line pressure gage on the vacuum line providing a means to quantify $CO_2$ (we discuss this in response to a later comment).

We propose to clarify this point within the manuscript:

"Following incubation, samples were acidified to pH 2 using DIC-free HCl. Evolved $CO_2$ from microbial respiration of DOC was sparged from solution (2 h, 0.08 L min$^{-1}$) with ultra-high purity He and trapped cryogenically (liquid $N_2$) on a vacuum line. The $CO_2$ was then purified through a series of cryogenic traps before being quantified and isolated in a pre-combusted (550°C, 5h) break seal (McCallister et al., 2006)."

*The experiment is lacking controls. For example, a killed control, or glacial water than remains acidified and uninoculated and is allowed to sit for the 28 days. How can you be sure that atmospheric CO2 was not introduced during the incubation, processing and harvesting, thereby skewing the value with modern CO2. It does not look that UHP He or O2 was used, nor were traps added to potentially trap any residual CO2 in these sparge gases which also might contribute to the recovered CO2 signature. Quantitative recovery is critical to ensure isotopic fidelity. Documented quantitative recovery could be accomplished by adding inv DIC with of a known quantity and isotopic signature and measuring the percent recovery and isotopic signature of the product. Currently it is impossible evaluate the validity of the reported respiratory CO2 isotopic signatures.*

Thank you for your questions. Firstly, ultra-high purity gases were used in all cases, and we propose to clarify this in text. We appreciate the Reviewer's desire to include a control but given the requirement of sufficient DIC production (i.e., active metabolism of organics) for the RCRS method we cannot send a control sample for isotopic analysis as when undertaken there are no organics metabolized, DIC produced and thus $CO_2$ to capture. This alone confirms that no gases are introduced into the gas-tight bottles or through the vacuum line. To further clarify, all incubation containers were gas tight, using septa that are routinely used for the study of DIC and gases such as methane (https://chemglass.com/bottles-anaerobic-media-clear). Given their frequent use and testing, we have no reason to believe that atmospheric gases would be introduced into the gas-tight incubation bottles. We have also

prepared a DIC standard, acidified it to pH 2 and captured the evolved $CO_2$ on the vacuum lines. Given the same concentration between the DIC in solution and captured $CO_2$, there is no evidence to suggest that there are leaks or fractionation issues associated with the lines. The vacuum lines are also under vacuum and monitored at numerous sites by gauges, so any leak would also be obvious. We sincerely hope that this testing and additional insight into the methodology has alleviated their concerns. Additionally, this is an established methodology (Guillemette, McCallister, & del Giorgio, 2013, 2016; McCallister & del Giorgio, 2008, 2012; McCallister et al., 2006), if not somewhat seldom utilized, where previous works have highlighted the potential for methodological contamination and fractionation artifacts to be negligible.

Finally, we view the samples analyzed here, which have similar starting DOC concentrations, isotopic signatures and DOM composition as pseudo-field replicates. Given the associated costs and time-consuming nature of the methodology, field replication was more beneficial for beginning to provide a complete understanding of processes driving bioavailability across glaciers. For most analyses involving radiocarbon in other settings (e.g. sediments, soils and DOC) it is typical not to analyze replicates. Therefore, we chose not to do replicate experiments and measurements of the same glacier meltwaters but encourage the Reviewer and readers of this manuscript to view the samples themselves as replicates within the glacier environment. With this in mind, we have reported ranges and averages throughout and have been careful not to put too much emphasis on singular isotopic or compositional values. Thus, we hope this highlights the uncertainty associated with this study and its aim to provide a first quantitative glimpse into the source(s) driving DOC bioavailability.

*Results and discussion: What amount of respiratory CO2 was captured for each incubation? Using that number, the percentage of bioavailable OC could be calculated. How does this compare to past published results? One can assume on 28-day timescales that BGE is low and that the majority of DOC consumed is fueling respiration rather than biomass production. But the authors should be careful to note this in their discussion, and elsewhere..that the portion that is respired is only a portion (though likely the majority) of bioavailable OC.*

We thank the Reviewer for their comment, between 0.2 - 0.3 mg C were captured during the incubations. Evolved $CO_2$ was quantified on the vacuum line, which has an inline pressure gage that was previously calibrated to known $CO_2$ quantities (spanning the range recovered here). We propose to report these quantities in the text and within the supplemental materials. Although we could in theory calculate % bioavailable DOC (BDOC) values based on evolved $CO_2$, we do not want to mislead the reader. As the Reviewer notes, classic %BDOC measurements are done by comparing DOC concentrations before and after incubation (e.g., Fellman et al., 2010; Hood et al., 2009). In this way, they consider DOC removal due to both biomass production and respiration. The %BDOC values that we could calculate here are using very different methods (i.e., altered geochemical conditions, inocula rather than in situ community, no filtration and direct acidification to end incubations, and indirect DIC quantification rather than DOC quantification) and only consider respiration. Thus, we do not believe these values should be compared to those in the literature directly, although we have provided the quantities in case individual readers are curious.

We appreciate the Reviewers point surrounding the allocation of DOC to respiration versus biomass production and their respective contributions to bioavailability. We have been careful throughout to refer to the respired OC or $CO_2$ produced by respiration. Nonetheless, we suggest a number of changes to further clarify this point:

1. In the title change 'utilize' to 'respire'
2. Highlighting that it is the respiratory fraction rather than 'bioavailable' in plain language summary: "Using bioincubation experiments we show glacier DOC **respiration**…"

3. In the study aims highlight respiratory rather than total bioavailable fraction assessed: "Here we investigate the age and potential sources of the **respired** fraction of DOC in a supraglacial stream and three glacier outflows in the Alaska Coast Mountains (Figure 1)."

4. Add clarifying sentence to the paragraph discussing study limitations as follows: "Moreover, microbial carbon utilization and DOC bioavailability in the environment are a function of both respiration and biomass production, the latter of which was not assessed here."

*Lines 157-158- I would indicate that this is a two-endmember model and that it is assumed that all of the radiocarbon dead material is derived from fossil fuel combustion byproducts.*

Thank you, we suggest the following edit: "Based on simple mixing of radiocarbon dead (**i.e., a purely fossil fuel source**) and modern OC, ~55 – 74 % of DOC across our study sites could have been derived from fossil fuel combustion byproducts, in line with past estimates (Table 1; Stubbins et al., 2012)."

*Line 184-186 and Figure 4- The trend of enriched d13C and younger CO2 with % RA of aliphatic compounds is driven largely by one point (supraglacial stream) out of 4. I think these statements should be better qualified to recognize the limitations in the dataset.*

Thank you for your comment. The supraglacial stream does have more enriched $\delta^{13}C$-$CO_2$ values than the outflows. However, even when removing this sample from the analyses, we find that increased aliphatic content (%RA) of DOM does tend to lead to $^{13}C$ enriched values of $CO_2$. Given the small number of samples, we have chosen not to quantify this shift, or statistically evaluate the trends, as to not mislead reader. We feel this is the appropriate caveat for the data presented in Figure 4. Although small in size, this data is still interesting, and we hope that it acts as a first step in quantifying the drivers of bioavailability and organic carbon respiration in glacier ecosystems as we have noted in text (e.g., lines 238).

We suggest further refinements to the sentences discussing aliphatic content and $\delta^{13}C$-$CO_2$ in order to not mislead the reader:

1. Remove the sentence discussing molecular data from the abstract.
2. Edit the results and discussion as follows:
   a. "Enriched $\delta^{13}C$-$CO_2$ values may be associated with an increased %RA of aliphatic compounds in the initial DOM pool, and in the case of the supraglacial stream appear to coincide with substantially younger $CO_2$ (Figure 4). Given the small dataset ($n = 4$), it is unclear whether this represents a compositional trend. However, the data suggest that the $^{13}C$ and $^{14}C$ enriched source respired during the incubations could contain a greater relative abundance of aliphatic compounds."
   b. "Furthermore, the supraglacial stream had the most aliphatic-rich DOM and the most $^{13}C$ and $^{14}C$ enriched $CO_2$ (Figure 2, 3 and 4). This is consistent with autochthonously sourced aliphatic organic matter (Musilova et al., 2017), and further suggests that respired organics were derived from in situ production."

*Line 186-187-I disagree with the authors' interpretation "This tendency suggests the 13C and 14C enriched source respired during the incubations may be relatively more abundant in the more aliphatic samples." Instead doesn't*

*this tendency suggest that the 13C and 14C enriched source which is respired contains a greater percentage of aliphatic OC?*

We agree with your interpretation. We suggest editing to: "However, the data suggest that the $^{13}$C and $^{14}$C enriched OC source respired during the incubation could contain a greater relative abundance of aliphatic compounds."

*Line 197- The significant figures for d13C here and else where in the manuscript should be consistent and likely to the second decimal place, but to the first decimal place at a minimum.*

We thank the Reviewer for their comment. We have ensured the significant figures on measured values are consistent in the text and tables. For isotope values, we report to one decimal place. The discrepancy noted by the Reviewer is with those reported from the literature, where the range of endmember values is so large as not to warrant the additional decimal places.

*Line 200-201- Change "were postulated to reflect greater in situ produced over atmospherically deposited OC (Schmidt et al., 2022)" to "were postulated to reflect a greater contribution of in situ produced OC relative to atmospherically deposited OC" or something similar to clarify the meaning.*

We thank you for the suggestion. The text would read:

"Similarly, total OC concentrations of cryoconite hole sediments have been shown to positively correlate with $\delta^{13}$C, where enriched values (~-16 to -10 ‰) **were postulated to reflect greater relative contributions of in situ produced OC compared to atmospherically deposited OC (Schmidt et al., 2022)."**

*Line 206-211- "Additionally, we observe relative similarity in molecular and isotopic composition (DOC and respiratory CO2) between the supraglacial stream and glacier outflows, supporting a component of the bioavailable organics exported from these glaciers being derived from recent in situ production on the surface, rather than a subglacial source (i.e., since the outflow and supraglacial samples are relatively comparable there is little evidence for significant subglacial input of OC; Table 1, Figure 2 & 3). This sentence needs to be more concise or broken into two sentences.*

Thank you, we propose clarifying to:

"Additionally, the relative similarity in molecular and isotopic composition (DOC and respiratory CO$_2$) between the supraglacial stream and the glacier outflow samples suggests that a portion of the bioavailable organic carbon exported from these glaciers originates from recent in situ production on the surface rather than subglacial sources (Table 1, Figures 2 & 3). However, subglacial sources such as paleosols and microbial chemotrophic processes could still contribute organic carbon to the outflow DOM pool."

*Line 225-226- Please clarify "the observed 13C-CO2 enrichment could have stemmed from a 13C enriched fossil fuel source or compounds within the sources OC pool, or result from photodegradation processes (Spencer et al., 2009)". I am confused as to how to reconcile the isotopic differences based on this explanation.*

Thank you, we propose to clarify the sentence to:

"Fossil fuel sources are on average more $^{13}C$ depleted compared to the isotopic composition of respiratory $CO_2$ (Table 1; Wang et al., 2022). Therefore, if fossil fuels contribute $^{13}C$ enriched OC, this could have originated from a $^{13}C$ enriched fossil fuel source, compounds within the sources' OC pool that are relatively $^{13}C$ enriched compared to the bulk, or result from photodegradation of fossil fuel-derived compounds which is known to result in $^{13}C$ enrichment (Spencer et al., 2009)."

*Table 1 lacks uncertainties for all measurements. Duplicates for DOC and d13C could have been easily run. In addition, for the D14C respiration measurements, we have no assessment of error or reproducibility. I recognize the cost of D14C measurements, but at least one of the samples should have been replicated to provide an error associated with the respiratory CO2 measurement. Further, what is the blank associated with the measurement? The introduction of atmospheric CO2 during the incubation and CO2 harvest could have easily shifted the respiratory CO2 to the more modern and d13C enriched (~ -7 per mil) values. Without a process blank one cannot back out isotopic artefacts from the experimental treatment. In addition, there are no CO2 recovery values. These amounts could be used to compare recoveries to previous C consumption in bioassays to see if they are realistic. They could also be employed to run some back of the envelope calculations on the bulk DOC to see whether it is feasible for x portion of a C13 enriched pool to be hidden.*

Thank you for your comment. We have discussed some of this above particularly with respect to controls and replicates. To further aid transparency in line with the Reviewer's comments, we propose to add the following details to the methods to clarify uncertainty:

"Concentrations of DOC were measured on a Shimadzu TOC-L$_{CPH}$ analyzer following standard methods (Holt et al., 2023 and references therein). Before analysis, samples were sparged with air for five minutes at a flow rate of 0.08 L min$^{-1}$ to remove DIC from solution. **Measured concentrations are based on 3 of 7 replicate injections with a coefficient of variance of <2%.** $^{13}C$ and $^{14}C$ were measured via isotope ratio mass spectrometry (IRMS) and accelerator mass spectrometry, respectively at Woods Hole Oceanographic Institution. For DOC isotopes, samples were UV-oxidized, and the resultant $CO_2$ cryogenically trapped for analysis. **$\delta^{13}C$ values measured by IRMS have a typical precision of <0.2‰ (Xu et al., 2021).** Estimates of the contributions from radiocarbon dead (-1000 ‰) versus modern (95 % of $^{14}C$ concentration in 1950 of NBS Oxalic Acid I normalized to $\delta^{13}C_{VPDB}$= -19 ‰) OC were calculated from fraction modern ($F_m$) values (Table 1), where the percentage of radiocarbon dead material was determined as 1-$F_m$ (Stubbins et al., 2012). **Measurement error on $F_m$ values ranged from 0.0017 to 0.0025 (Table 1), making little quantitative difference to calculated values (i.e., ‰ and yBP) and estimated source contributions."**

We also propose adding error on $F_m$ values to Table 1.

As explained in previous responses, RCRS incubations were conducted in gas tight bottles using septa typically used in DIC and methane analyses, including bioincubations. This gas tight set up has been used in multiple studies across decades of research and thus we have no reason to think they would not be gas tight in our case. Furthermore, quantitative recovery of a DIC standard and a control sample (which measured no $CO_2$ recovery on the vacuum line), indicates that introduction of atmospheric gases during sample incubation, extraction and cryogenic purification is not an issue within the RCRS methodology. Finally, as noted above, the lines are under vacuum and monitored at numerous locations, so introduction of any atmospheric gases would be readily apparent.

With regards to duplicate incubations, we understand the Reviewer's desire for some quality control but, unfortunately, an $n$ of 2 would really only tell you if they were different and so we would need an $n$ of 3 at least to begin to get at reproducibility. As noted above, this is just not typically undertaken in methodology involving radiocarbon analyses due to prohibitive costs. Instead, we opted for pseudo-field replication of compositionally similar glacier meltwaters ($n = 4$, supraglacial and outflows). We hoped this would provide a broader range of isotopic values and thus sources than could be achieved with experimental replication, enabling a more complete understanding of drivers of bioavailability. Importantly, across all glacier samples, younger and $^{13}C$ enriched organics were preferentially respired relative to bulk DOC. This aligns with real-world expectations and is consistent with the current literature. For example, global molecular-level assessment of glacier outflow DOM composition indicates an increased %RA of aliphatic compounds in the DOM pool is associated with younger and $^{13}C$-enriched DOC, suggesting bioavailable organics are derived from a modern, $^{13}C$ enriched, in situ-produced source (Holt et al., 2023; Holt et al., 2024). This is also in agreement with numerous studies that indicate in situ production on glacier surfaces is a source of biolabile DOC (e.g., Musilova et al., 2017; Smith et al., 2017).

Finally, as explained above, we propose to report $CO_2$ quantities sent for isotopic analysis in the text and supplementary materials in the revised manuscript. However, given the significant differences between the RCRS method and classic BDOC incubations, we do not think it is appropriate to calculate %BDOC values here since we do not wish to mislead or confuse the reader. If studies do so in the future, we recommend quantitative recoveries of $CO_2$ should first be compared directly to the traditional BDOC incubation method to validate this approach, something we unfortunately did not do as part of this study.

We thank you for your helpful comments on this manuscript, we hope, if the changes outlined here are adopted, you feel it is now suitable for publication.

Yours Sincerely,

Amy D. Holt

References:

Andrews, M. G., Jacobson, A. D., Osburn, M. R., & Flynn, T. M. (2018). Dissolved carbon dynamics in meltwaters from the Russell Glacier, Greenland Ice Sheet. *Journal of Geophysical Research: Biogeosciences, 123*(9), 2922-2940.

Fellman, J. B., Spencer, R. G., Hernes, P. J., Edwards, R. T., D'Amore, D. V., & Hood, E. (2010). The impact of glacier runoff on the biodegradability and biochemical composition of terrigenous dissolved organic matter in near-shore marine ecosystems. *Marine Chemistry, 121*(1-4), 112-122.

Guillemette, F., McCallister, S. L., & del Giorgio, P. A. (2013). Differentiating the degradation dynamics of algal and terrestrial carbon within complex natural dissolved organic carbon in temperate lakes. *Journal of Geophysical Research: Biogeosciences, 118*(3), 963-973.

Guillemette, F., McCallister, S. L., & Del Giorgio, P. A. (2016). Selective consumption and metabolic allocation of terrestrial and algal carbon determine allochthony in lake bacteria. *The ISME journal, 10*(6), 1373-1382.

Harley, J. R., Biles, F. E., Brooks, M. K., Fellman, J., Hood, E., & D'Amore, D. V. (2023). Riverine dissolved inorganic carbon export from the Southeast Alaskan Drainage Basin with implications for coastal ocean processes. *Journal of Geophysical Research: Biogeosciences, 128*(10), e2023JG007609.

Holt, A. D., Kellerman, A. M., Battin, T. I., McKenna, A. M., Hood, E., Andino, P., . . . De Staercke, V. (2023). A tropical cocktail of organic matter sources: Variability in supraglacial and glacier outflow dissolved organic matter composition and age across the Ecuadorian Andes. *Journal of Geophysical Research: Biogeosciences*, e2022JG007188.

Holt, A. D., McKenna, A. M., Kellerman, A. M., Battin, T. I., Fellman, J. B., Hood, E., . . . Styllas, M. (2024). Gradients of deposition and in situ production drive global glacier organic matter composition. *Global Biogeochemical Cycles, 38*(9), e2024GB008212.

Hood, E., Fellman, J., Spencer, R., Hernes, P., Edwards, R., D'Amore, D., & Scott, D. (2009). Glaciers as a source of ancient and labile organic matter to the marine environment. *Letters to Nature, 462*(24), 1044-1048. doi:doi:10.1038/nature08580

Kellerman, A. M., Vonk, J., McColaugh, S., Podgorski, D. C., van Winden, E., Hawkings, J. R., . . . Spencer, R. G. (2021). Molecular signatures of glacial dissolved organic matter from Svalbard and Greenland. *Global Biogeochemical Cycles, 35*(3), e2020GB006709.

McCallister, S. L., & del Giorgio, P. A. (2008). Direct measurement of the d13C signature of carbon respired by bacteria in lakes: Linkages to potential carbon sources, ecosystem baseline metabolism, and CO2 fluxes. *Limnology and Oceanography, 53*(4), 1204-1216.

McCallister, S. L., & del Giorgio, P. A. (2012). Evidence for the respiration of ancient terrestrial organic C in northern temperate lakes and streams. *Proceedings of the National Academy of Sciences, 109*(42), 16963-16968.

McCallister, S. L., Guillemette, F., & Del Giorgio, P. A. (2006). A system to quantitatively recover bacterioplankton respiratory CO2 for isotopic analysis to trace sources and ages of organic matter consumed in freshwaters. *Limnology and Oceanography: Methods, 4*(10), 406-415.

Musilova, M., Tranter, M., Wadham, J., Telling, J., Tedstone, A., & Anesio, A. M. (2017). Microbially driven export of labile organic carbon from the Greenland ice sheet. *Nature Geoscience, 10*(5), 360-360.

Schmidt, S. K., Johnson, B. W., Solon, A. J., Sommers, P., Darcy, J. L., Vincent, K., . . . Porazinska, D. L. (2022). Microbial biogeochemistry and phosphorus limitation in cryoconite holes on glaciers across the Taylor Valley, McMurdo Dry Valleys, Antarctica. *Biogeochemistry, 158*(3), 313-326.

Smith, H. J., Foster, R. A., McKnight, D. M., Lisle, J. T., Littmann, S., Kuypers, M. M., & Foreman, C. M. (2017). Microbial formation of labile organic carbon in Antarctic glacial environments. *Nature Geoscience, 10*(5), 356-359.

Spencer, R. G., Guo, W., Raymond, P. A., Dittmar, T., Hood, E., Fellman, J., & Stubbins, A. (2014). Source and biolability of ancient dissolved organic matter in glacier and lake ecosystems on the Tibetan Plateau. *Geochimica et Cosmochimica Acta, 142*, 64-74.

Spencer, R. G. M., Stubbins, A., Hernes, P. J., Baker, A., Mopper, K., Aufdenkampe, A. K., . . . Wabakanghanzi, J. N. (2009). Photochemical degradation of dissolved organic matter and dissolved lignin phenols from the Congo River. *Journal of Geophysical Research: Biogeosciences, 114*(G3).

Stubbins, A., & Dittmar, T. (2012). Low volume quantification of dissolved organic carbon and dissolved nitrogen. *Limnology and Oceanography: Methods, 10*(5), 347-352.

Wang, P., Zhou, W., Xiong, X., Wu, S., Niu, Z., Cheng, P., . . . Hou, Y. (2022). Stable carbon isotopic characteristics of fossil fuels in China. *Science of The Total Environment, 805*, 150240.

---

## Author Comment (AC2)

**FLORIDA STATE UNIVERSITY**

College of Arts & Sciences
*Department of Earth, Ocean & Atmospheric Science*

April 2nd, 2025

Dr. Amy Holt

Earth, Ocean, and Atmospheric Science (EOAS)

Florida State University

Tallahassee, FL 32306-4520

Dear Reviewer 2,

**Brief Communications: Stream Microbes Preferentially Utilize Young Carbon within the Ancient Bulk Glacier Dissolved Organic Carbon Pool.**

Amy D. Holt, Jason B. Fellman, Anne M. Kellerman, Eran Hood, Samantha H. Bosman, Amy M. McKenna, Jeffery P. Chanton & Robert G. M. Spencer.

We sincerely thank you for your positive feedback and helpful comments on our manuscript. Please find below a detailed response to your comments with proposed revisions. Your comments are in italic and our response in regular text. Line numbers refer to the original posted manuscript. We hope with these refinements and additions you now find the paper suitable for publication in The Cryosphere: Brief Communications.

**Response:**

*General comments: This paper uses multiple analytical techniques to characterize the pool of dissolved organic carbon in the outflow of three Alaskan Glaciers, and in a supraglacial stream of one of those glaciers. The authors pair these measurements of glacial DOC with bioincubation experiments and measure the isotopic signature of respired $CO_2$. They conclude that the young and aliphatic-rich portion of the DOM pool is most bioavailable. The paper is generally well written, and the suite of measurements provide valuable insights into glacially-sourced DOM cycling, which is of appropriate scope for The Cryosphere. However, there are questions/comments detailed below regarding conclusions drawn from this experimental design, which lacks controls, replicates, and measures of uncertainty. These issues should be addressed, and conclusions carefully articulated within the limitations of the dataset.*

We thank the Reviewer for their positive feedback, we have addressed your concerns below.

*Specific Comments: Figure 1: this figure could be significantly improved so that each glacier is visible. Consider using a satellite image (from Jul, or ideally close to the field sampling period) and zoom in enough so that precise sampling locations are visible relative to the glacier terminus/proglacial lake/stream network.*

We thank the Reviewer for the advice and suggestion. We would like to change the map to that below:

[Figure]

**Figure 1:** Location of July 2022 sample sites within (a.) the Juneau Icefield highlighting panels (b.-d.), and (b.) Mendenhall Glacier, (c.) Herbert Glacier and Eagle Glacier catchments. Map data: Bing Satellite.

*Section 2.1: This section would benefit from a more detailed description of these glaciers, including comments on their potential subglacial hydrology and geology, and biogeochemical context from previous work.*

Thank you for the suggestion. We propose the following edits and additions to the site description:

"Water samples were collected between the 11th and 14th of July 2022, from three glacierized watersheds in coastal southeast Alaska (Figure 1). The study area is situated within the Juneau Icefield, in the coastal temperate rainforest and has a cool (annual mean temperature 5.6°C), maritime climate, with the majority of precipitation falling in autumn and winter (Behnke et al., 2020). The geology of the upper watersheds, where the glaciers are found, is dominated by Tertiary-Cretaceous aged, foliated tonalite sill of the coast plutonic complex (Wilson et al., 2015). These glaciers are well studied and are known to have discharge regimes and biogeochemical characteristics representative of glacial systems throughout the Gulf of Alaska. These glacier rivers are highly turbid, with low summer temperatures (<5°C) and oligotrophic conditions (Hood and Scott, 2008; Fellman et al., 2014; Spencer et al., 2014). Based on past observations of conductivity, at the time of sampling, water residence times within the glaciers are short (~hours), and subglacial drainage is relatively efficient, with the vast majority of meltwater having a supraglacial origin (Spencer et al., 2014).

Water samples were collected from the surface and outflow of Mendenhall Glacier, as well as downstream (~ ≤1 km) of the terminus of both Eagle and Herbert Glaciers. Between the glacier terminus and sampling sites glacier outflow rivers flowed through recently deglaciated terrain (i.e., barren ground of cobble, gravel and glacier silt, with few colonizer plants) and thus there was limited potential for OC inputs from vascular plants and soil organic matter. At Mendenhall, outflow sampling was conducted on a rock/silt bar <100 m from the glacier outflow. Water here is extremely turbulent and flows rapidly into Mendenhall lake. As such, there is limited influence of the lake water, and sampling is representative of water exiting the glacier. The supraglacial sample was collected from a small (<1 m across) flowing supraglacial stream on the bare ice surface accessed by helicopter, ~ 3 km upslope from the glacier terminus."

*Line 77: provide more details on where the supraglacial sample was collected. Improvements to Figure 1 will help, but it's worth noting here that it came from a supraglacial stream (which you mention in results). Knowing where was collected relative to the terminus/snow line has implications for interpreting DOC sources.*

Thank you and in line with your previous comment, we suggest adding the following to text:

"The supraglacial sample was collected from a small (<1 m across) flowing supraglacial stream on the bare ice surface accessed by helicopter, ~ 3 km upslope from the glacier terminus."

*Line 77: describe where you collected your outflow samples in each catchment. This is especially important for Mendenhall since it terminates in a proglacial lake. Understanding the location relative to local/micro-environments and non-glacial stream influence is relevant to water/DOC source, and residence/transit times. This would also be a good spot to describe the timing of your sample collection in relation to the seasonal evolution of the glacial hydrology and ultimately theoretical water/DOM sources. Do you expect the outflow at this time of year to represent an efficient drainage system with little influence from subglacial sources? If so, that means this paper is not really looking at the 'bulk glacier DOM pool' as the title suggests, and is instead looking at the lability of predominantly supraglacial DOM.*

Thank you for the suggestion. As described above, we propose to add details of sampling location and glacier hydrology to text. We also suggest changing 'streamwater' to 'water' in the methods and throughout, in order to not mislead the reader as to the nature of the water samples at Mendenhall.

To clarify the meaning of 'bulk' here, this refers to the total DOC pool of samples, in order to distinguish it from the fractionated or respired component. As indicated by the Reviewer, we do intend to refer to the typical composition or all the possible compositions of supraglacial and outflow DOM. We suggest removing bulk from the title and from line 36 where the meaning may be confused. Nonetheless, given our sampling was focused on months of peak melt, we did capture DOM composition at the peak of the DOM flux from these glacier environments. In this way, sampling was designed to be representative of the period of time where the majority of DOM and bioavailable DOC export occurs. As such, we do maintain that these findings are broadly representative of these glacier outflows, although temporal variability could affect OC sources on a seasonal or annual basis (as acknowledged in text lines 233-240).

*Table 1: report an indicator of uncertainty (including detection limits) for these analyses, here or elsewhere in the document. This is particularly important for the instances where you compare supraglacial DOM to outflow DOM.*

Thank you for your comment. We propose to add the following details to the methods to clarify uncertainty:

"Concentrations of DOC were measured on a Shimadzu TOC-L$_{CPH}$ analyzer following standard methods (Holt et al., 2023 and references therein). Before analysis, samples were sparged with air for five minutes at a flow rate of 0.08 L min$^{-1}$ to remove DIC from solution. **Measured concentrations are based on 3 of 7 replicate injections with a coefficient of variance of <2%.** $^{13}$C and $^{14}$C were measured via isotope ratio mass spectrometry (IRMS) and accelerator mass spectrometry, respectively at Woods Hole Oceanographic Institution. For DOC isotopes, samples were UV-oxidized, and the resultant $CO_2$ cryogenically trapped for analysis. **δ$^{13}$C values measured by IRMS have a typical precision of <0.2‰ (Xu et al., 2021).** Estimates of the contributions from radiocarbon dead (-1000 ‰) versus modern (95 % of $^{14}$C concentration in 1950 of NBS Oxalic Acid I normalized to δ$^{13}$C$_{VPDB}$= -19 ‰) OC were calculated from fraction modern (F$_m$) values (Table 1), where the percentage of radiocarbon dead material was determined as 1-F$_m$ (Stubbins et al., 2012). **Measurement error on F$_m$ values ranged from 0.0017 to 0.0025 (Table 1), making little quantitative difference to calculated values (i.e., ‰ and yBP) and estimated source contributions.**"

To clarify, the comparisons made between the surface and outflow composition are appropriate given these uncertainty levels. We maintain that molecular-level composition is relatively similar. For example, the relative abundance (RA) of aliphatic compounds between samples ranges from 21.0 – 34.2%. For comparison, past studies of global glacier outflows demonstrate this range is 8.6 – 71.5 %RA (Holt et al., 2024). Similarly, the isotopic composition of supraglacial respired OC was far younger (-30.8 compared to -146.2 – 232.9) and $^{13}$C enriched (-15.5 compared to -18.3 - 19.5 ‰; Table 1), considering the uncertainty of measured values.

*Figure 2: (a) and (b) are unnecessary since the same data are presented more efficiently in (c)*

Thank you for the suggestion, we propose to remove panels a and b. We simply presented them in this way originally to aid accessibility and introduce stable and radiocarbon isotope data independently, but we are happy to remove them.

*Section 3.2, line 157, line 210: given the increasing evidence for subglacial chemotrophic pathways, it's worth commenting on the likelihood of these sources of DOM to the outflows, especially whether the subglacial environment may be a source of old but labile DOM. Additional information on the subglacial hydrology/geology in section 2.1 will help frame whether these systems are conductive to chemotrophic DOM production. Are there previous studies at your sites that would justify ruling this source out? Perhaps this is what you're referring to in line 228, but I think it warrants a more thorough discussion than the statement in brackets on line 209-210.*

Thank you for your comment. Previous studies at these sites have found no evidence for subglacial inputs at this time of year (peak melt), as indicated by similarity in isotopic signature and DOC concentrations and composition, and between the surface and outflow (Spencer et al., 2014; Stubbins et al., 2012). As explained in the text, we also note broad similarity in molecular and isotopic composition between outflows and the surface sample, indicating minimal inputs from aged subglacial sources and likely a predominately supraglacial origin of biolabile organics at the time of sampling. Although we cannot, and do not, rule out the possibility of some aged material originating from the subglacial environment, more research is required to validate this as a bioavailable carbon source at this location.

To reiterate, we note the dominant source of respired OC is young and $^{13}$C enriched, with minimal inputs from aged OC regardless of origin (i.e., supra- or subglacial). We have noted the potential for subglacial OC contributing

organics within the manuscript and this was meant to encompass subglacial microbial processes. Nonetheless, we propose to further clarify this as suggested below:

1. "Additionally, the relative similarity in molecular and isotopic composition (DOC and respiratory $CO_2$) between the supraglacial stream and the glacier outflow samples suggests that a portion of the bioavailable organic carbon exported from these glaciers originates from recent in situ production on the surface rather than subglacial sources (Table 1, Figures 2 & 3). However, subglacial sources such as paleosols and microbial chemotrophic processes could still contribute organic carbon to the outflow DOM pool."
2. "In outflows, ancient DOC compounds could also be derived from aged ice-locked, and subglacial sources (e.g., paleosols and microbial chemotrophic processes), although compositional and carbon isotopic evidence to support this conclusively is yet to be observed for Alaskan glaciers…."

*Line 184-187: Without error bars or statistical analysis (Figure 4), I'm not sure you can claim these associations are significant or therefore make a justified link between the 13C and 14C signatures to the aliphatic content. Further, since the DOM characterization is not truly quantitative, RA was not measured before and after the incubation, and you do not present CO2 concentrations or quantify the DOC consumed during the incubation, I think it's a stretch to claim that "the 13C and 14C enriched source respired during the incubation may be relatively more abundant in the more aliphatic samples" and then extend that further in the abstract to say "Molecular-level analyses suggest respired OC was associated with the aliphatic-rich portion of the DOM pool".*

Thank you for the comment. To clarify, we do not make the case that these trends are significant. Given the small number of samples, we agree statistical analysis is not appropriate and thus we cannot make the claim that the trends are significant. As explained in response to Reviewer 1, we have deliberately chosen not to quantify this shift, or statistically evaluate the trends, as to not mislead reader. This is why there are no regression lines or statistical analyses presented in Figure 4 or the text. We feel this is the appropriate caveat for the data presented in Figure 4. Although small in size, this data remains very interesting, and we hope that it acts as a first step in quantifying the drivers of bioavailability and organic carbon respiration in glacier ecosystems as we have noted in text (e.g., lines 238 onwards). This is one of the reasons we think this is a great dataset for a Brief Communications article to get this into the community, stimulating more research on this topic.

To make this clearer as suggested, we propose further refinements to the sentences discussing aliphatic content and $\delta^{13}C$-$CO_2$ in order to not mislead the reader:

1. Remove the sentence discussing molecular data from the abstract.
2. Edit the results and discussion as follows:
    a. Enriched $\delta^{13}C$-$CO_2$ values may be associated with an increased %RA of aliphatic compounds in the initial DOM pool, and in the case of the supraglacial stream appear to coincide with substantially younger $CO_2$ (Figure 4). Given the small dataset (n = 4), it is unclear whether this represents a compositional trend. However, the data suggest that the $^{13}C$ and $^{14}C$ enriched source respired during the incubations could contain a greater relative abundance of aliphatic compounds."
    b. "Furthermore, the supraglacial stream had the most aliphatic-rich DOM and the most $^{13}C$ and $^{14}C$ enriched $CO_2$ (Figure 2, 3 and 4). This is consistent with autochthonously sourced aliphatic organic matter (Musilova et al., 2017), and further suggests that respired organics were derived from in situ production."

*Section 3.4: Do you know that there were no inorganic processes affecting CO2 concentrations/isotopic signatures, despite the 0.45 um filter? Glacial flour can have an ultra-fine component with high surface area, so it's theoretically possible. It appears you did not collect CO2 concentrations or DOC after incubation, which is a shame since that would allow for a simple mass balance calculation and help isolate/confirm the process(es) at play. An experimental control would have also addressed this issue. In the absence of these data and a robust experimental design with controls, I think you need to somehow make your case that inorganic processes and/or contamination are not contributing CO2 during incubation.*

Thank you for your question. To clarify, samples for RCRS bioincubation were collected, filtered (0.45 µm), acidified (pH 2) and stored frozen. This is common methodology for DOC and DOM analyses of glacier waters, despite the high turbidity and fine glacier flour in glacier outflows. We did not observe any glacier flour in the samples after filtration, collection or when samples were thawed. As noted to Reviewer 1, we also did not observe any flocculation. Samples for RCRS incubation were acidified and sparged to remove inorganic carbon (IC) from solution before the bioincubation was started. Therefore, any IC (including those within any glacial flour <0.45 µm) would also be removed. As explained in response to Reviewer 1, an experimental control does not produce DIC as there is no active metabolism of organics. Therefore, we cannot send anything for isotopic analysis as there is no DIC produced and thus no $CO_2$ to capture. Furthermore, the observed isotopic shifts for glacier outflows and supraglacial samples were similar (Figure 3). Given surface samples have negligible glacier flour, this further indicates that any potential glacier flour and its influence on IC is not an issue. Additionally, all incubation containers were gas tight, using septa that are routinely used for the study of DIC and gases such as methane. Given their frequent use and testing, we have no reason to believe that atmospheric gases would be introduced into the gas-tight incubation bottles. Finally, as also noted to Reviewer 1, there was good recovery on a DIC standard (i.e. same concentration of DIC in solution and captured $CO_2$) and the vacuum lines are kept under vacuum and so any contamination associated with the lines during extraction would also be obvious. Taken together, this highlights the data presented is in no way an artifact of methodological design. We sincerely hope this alleviates the Reviewer's concern.

We thank you for your helpful comments and time spent on this manuscript, we hope, if the changes outlined here are adopted, you feel it is now suitable for publication.

Yours Sincerely,

Amy D. Holt

References:

Behnke, M. I., Stubbins, A., Fellman, J. B., Hood, E., Dittmar, T., & Spencer, R. G. (2020). Dissolved organic matter sources in glacierized watersheds delineated through compositional and carbon isotopic modeling. *Limnology and Oceanography*.

Fellman, J. B., Nagorski, S., Pyare, S., Vermilyea, A. W., Scott, D., & Hood, E. (2014). Stream temperature response to variable glacier coverage in coastal watersheds of Southeast Alaska. *Hydrological Processes, 28*(4), 2062-2073.

Holt, A. D., Kellerman, A. M., Battin, T. I., McKenna, A. M., Hood, E., Andino, P., . . . De Staercke, V. (2023). A tropical cocktail of organic matter sources: Variability in supraglacial and glacier outflow dissolved organic matter composition and age across the Ecuadorian Andes. *Journal of Geophysical Research: Biogeosciences*, e2022JG007188.

Holt, A. D., McKenna, A. M., Kellerman, A. M., Battin, T. I., Fellman, J. B., Hood, E., . . . Styllas, M. (2024). Gradients of deposition and in situ production drive global glacier organic matter composition. *Global Biogeochemical Cycles, 38*(9), e2024GB008212.

Hood, E., & Scott, D. (2008). Riverine organic matter and nutrients in southeast Alaska affected by glacial coverage. *Nature Geoscience, 1*(9), 583-587.

Musilova, M., Tranter, M., Wadham, J., Telling, J., Tedstone, A., & Anesio, A. M. (2017). Microbially driven export of labile organic carbon from the Greenland ice sheet. *Nature Geoscience, 10*(5), 360-360.

Smith, H. J., Foster, R. A., McKnight, D. M., Lisle, J. T., Littmann, S., Kuypers, M. M., & Foreman, C. M. (2017). Microbial formation of labile organic carbon in Antarctic glacial environments. *Nature Geoscience, 10*(5), 356-359.

Spencer, R. G., Vermilyea, A., Fellman, J., Raymond, P., Stubbins, A., Scott, D., & Hood, E. (2014). Seasonal variability of organic matter composition in an Alaskan glacier outflow: Insights into glacier carbon sources. *Environmental Research Letters, 9*(5), 55005-55005. doi:10.1088/1748-9326/9/5/055005

Stubbins, A., Hood, E., Raymond, P. A., Aiken, G. R., Sleighter, R. L., Hernes, P. J., . . . Schuster, P. (2012). Anthropogenic aerosols as a source of ancient dissolved organic matter in glaciers. *Nature Geoscience, 5*(3), 198.

Wilson, F. H., Hults, C. P., Mull, C. G., & Karl, S. M. (2015). *Geologic map of Alaska*: US Department of the Interior, US Geological Survey Reston, VA.

Xu, L., Roberts, M. L., Elder, K. L., Kurz, M. D., McNichol, A. P., Reddy, C. M., . . . Hanke, U. M. (2021). Radiocarbon in Dissolved Organic Carbon by Uv Oxidation: Procedures and Blank Characterization at Nosams. *Radiocarbon, 63*(1), 357-374.

---

## Author Response (AR2)

**FLORIDA STATE UNIVERSITY**

College of Arts & Sciences

*Department of Earth, Ocean & Atmospheric Science*

*Phone: +1-850-645-0955 \*Fax: 850-644-4214 \*Email: adh19d@fsu.edu*

May 27th, 2025

Dr. Amy Holt

Earth, Ocean, and Atmospheric Science (EOAS)

Florida State University

Tallahassee, FL 32306-4520

Dear Dr. Bagshaw,

**Brief Communications: Stream Microbes Preferentially Respire Young Carbon within the Ancient Glacier Dissolved Organic Carbon Pool**

Amy D. Holt, Jason B. Fellman, Anne M. Kellerman, Eran Hood, Samantha H. Bosman, Amy M. McKenna, Jeffery P. Chanton & Robert G. M. Spencer

Thank you for the time spent reviewing our manuscript. In line with your suggestion, we have edited line 321 to read: *"In outflows, ancient DOC compounds could also be derived from aged ice-locked and subglacial sources (e.g., paleosols and microbial chemotrophic processes), although supporting data for these sources is yet to be observed for Alaskan glaciers (Stubbins et al., 2012; Spencer et al., 2014)."*

We are pleased you feel the manuscript is now ready for publication, and look forward to seeing it published in The Cryosphere Brief Communications.

Yours Sincerely,

Amy D. Holt